# Impact of the Liberian National Community Health Assistant Program on childhood illness care in Grand Bassa County, Liberia

Emily White[1], Savior Mendin[2], Featha R. Kolubah[3], Robert Karlay[2], Ben Grant[2], George P. Jacobs[4], Marion Subah[2], Mark J. Siedner[5,6], John D. Kraemer[7‡]*, Lisa R. Hirschhorn[8‡]

**1** Last Mile Health, Boston, Massachusetts, United States of America, **2** Last Mile Health, Monrovia, Liberia, **3** Grand Bassa County Health Team, Buchanan, Liberia, **4** Ministry of Health, Monrovia, Liberia, **5** Department of Medicine, Harvard Medical School, Boston, Massachusetts, United States of America, **6** Center for Global Health, Massachusetts General Hospital, Boston, Massachusetts, United States of America, **7** Department of Health Systems Administration, Georgetown University, Washington, D.C., United States of America, **8** Department of Medical Social Sciences, Northwestern University Feinberg School of Medicine, Chicago, Illinois, United States of America

☯ These authors contributed equally to this work.
‡ LRH and JDK also contributed equally as senior authors.
* jdk32@georgetown.edu

**Data Availability Statement:** Replication data are available at https://data.mendeley.com/datasets/7h9f9mkh9b/1. Statistical code to replicate

## Abstract

Liberia launched its National Community Health Assistant Program in 2016, which seeks to ensure that all people living 5 kilometers or farther from a health facility have access to trained, supplied, supervised, and paid community health workers (CHWs). This study aims to evaluate the impact of the national program following implementation in Grand Bassa County in 2018 using data from population-based surveys that included information on 1291 illness episodes. We measured before-to-after changes in care for childhood illness by qualified providers in a portion of the county that implemented in a first phase compared to those which had not yet implemented. We also assessed changes in whether children received oral rehydration therapy for diarrhea and malaria rapid diagnostic tests if they had a fever by a qualified provider (facility based or CHW). For these analyses, we used a difference-in-differences approach and adjusted for potential confounding using inverse probability of treatment weighting. We also assessed changes in the source from which care was received and examined changes by key dimensions of equity (distance from health facilities, maternal education, and household wealth). We found that care of childhood illness by a qualified provider increased by 60.3 percentage points (95%CI 44.7–76.0) more in intervention than comparison areas. Difference-in-differences for oral rehydration therapy and malaria rapid diagnostic tests were 37.6 (95%CI 19.5–55.8) and 38.5 (95%CI 19.9–57.0) percentage points, respectively. In intervention areas, care by a CHW increased from 0 to 81.6% and care from unqualified providers dropped. Increases in care by a qualified provider did not vary significantly by household wealth, remoteness, or maternal education. This evaluation found evidence that the Liberian National Community Health Assistant Program has increased access to effective care in rural Grand Bassa County. Improvements were approximately equal across three measured dimensions of marginalization.

analyses are included as a supplement to this manuscript.

**Funding:** This study was supported by the United States Agency for International Development's Development Innovation Ventures, the Margaret A. Cargill Foundation, and UBS Optimus Foundation. JDK receives salary support from Last Mile Health. Funders played no role in the design, conduct, or reporting of the study.

**Competing interests:** The authors have declared that no competing interests exist.

## Introduction

Liberia has made significant gains in child health since the end of its prolonged civil war. All-cause under-five mortality has reduced by approximately 15% since 2007 [1]. Nationwide, these improvements were driven by substantial investments in public sector delivery of health care, strengthening of facility-based services and improved public health services [2–4]. Despite these gains, Liberia continues to suffer one of the highest rates of child death in the world, and Liberian children who live in rural areas are about 20% more likely to die from any cause before their fifth birthday than their urban counterparts [1]. This disparity results in part from higher barriers to care for easily treatable childhood illnesses, including higher rates of poverty, insufficient access to transportation, delays from greater travel time, lower health literacy, and, in some instances, lower quality of health services in rural areas [5–7]. Because they provide services directly in rural communities, community health worker (CHW) programs can be well suited to overcoming these barriers [8–10].

In collaboration with the Liberian Ministry of Health, Last Mile Health, a non-governmental organization, implemented a pilot program of community health workers in Grand Gedeh County in 2012. The aim of the pilot program was to understand whether trained, well-supplied, and supervised CHWs could increase access to preventive and curative treatment for malaria, pneumonia and diarrhea and screening and referral for malnutrition among children and expand maternal and reproductive health in rural Liberian communities. Early evaluation of this pilot project demonstrated before-to-after improvements in management of these childhood illnesses and uptake in facility-based delivery [11]. This led to a modified demonstration project in Rivercess County, Liberia, which showed substantial increases in childhood treatment from qualified providers in a difference-in-differences analysis and that also showed that care shifted from informal and traditional providers toward CHWs [12].

Based in part on the early success of these initial pilot programs and lessons from the 2014–2015 Ebola epidemic response, Liberia adopted a National Community Health Assistant Program (NCHAP) in 2016 as part of a comprehensive strategy to extend the reach of the country's publicly operated primary health care system into areas more than 5km from a health facility. The program is led by the Ministry of Health and uses an integrated and standardized approach in which CHWs, called community health assistants (CHAs), are trained and supplied to deliver a package of interventions in their communities and systematically supervised to reinforce their training and skills [13, 14]. The program was informed by emerging evidence of CHW program design principles to leverage community-based care providers who can help strengthen primary care access and delivery through direct provision of evidence based interventions and linkage to facility based care responsive to their community health care needs, reducing barriers of access and cost and enabling increased timeliness and coverage of care [14, 15]. However, the impact of implementation under the full national program on access and uptake of health services has not been assessed. Further, more data are needed to assess whether the program has been implemented in a way that equitably includes the most vulnerable populations.

Our study seeks to evaluate the national CHA program implementation in rural Grand Bassa County, Liberia. The implementation was led by the Liberian Ministry of Health with technical and financial support from Last Mile Health in Grand Bassa County. (Other parts of the country had other non-governmental partners providing support to the health ministry.) Because program implementation was phased for operational reasons, we can assess program impact by measuring changes in child health care in areas that have already implemented compared to areas that have not implemented yet. This approach assumes that trends in the comparison area represent what would have happened in the intervention areas but for program

implementation. We aim to answer three questions. First, were there larger increases in child health care in intervention areas than in comparison areas where implementation has not yet begun? Second, if gains are observed, did the source of care shift to CHAs and facility-based providers from less effective informal sector providers? Third, how did changes in child health care in implementation areas vary across three measures of inequity: relative poverty, geographic remoteness, and maternal education?

## Methods

### Setting, population, and services

Grand Bassa is a predominately rural county in south-central Liberia, with a population of approximately 220,000 people spread over approximately 8000 square kilometers who are served by 31 health facilities [16]. It is divided into eight health districts. The CHA intervention began in Grand Bassa County in June 2018 and provides services to households in communities more than 5 km from the nearest health facility. Because of resource constraints, the program rollout was planned in four phases, with two health districts per phase. In this paper, we report results from the first phase of implementation.

Through the national program, CHAs provide a comprehensive package of health promotion and treatment services, including modules focused on maternal, newborn, and young child health; community registration; surveillance for diseases of epidemic potential; and family planning. These community-based providers are trained on integrated community case management protocols and provided commodities to treat uncomplicated suspected malaria, acute respiratory illness, and diarrhea and to support referral of complicated cases to health facilities. CHAs receive a monthly stipend of $70 USD and are supervised by Community Health Services Supervisors who are mostly nurses, physician assistants, or midwives. The CHSS conducts supervision visits to each community health assistant twice monthly. There were 100 CHAs and 10 Community Health Services Supervisors at the time of the evaluation. All services were delivered following the standard of care under Liberian policy. Further details about CHA recruitment, training, supervision, incentives, and provision have been previously reported [13–15].

### Data collection and sampling

We conducted two population-based, stratified cluster-sample surveys in February-June 2018 (pre-intervention) and February-April 2019 (5 months after phase 1 implementation). The baseline survey included all eight districts in Grand Bassa but our analytical sample is limited to the four districts sampled at follow-up; the follow-up survey included the two implementation districts and the two non-implementation districts that were most comparable on baseline child health outcomes. Prior to the baseline survey, we constructed a sampling frame by mapping all communities in the county and recording all households. The baseline sampling consisted of a simple random sample of communities, stratified by district, and selection of all households within each selected community. At follow-up, we sampled communities with probability of selection proportional to estimated size, stratified by implementation phase, and then selected 24 households per selected community by a modified random walk procedure. All women aged 18–49 in selected households were invited to complete the maternal and child health modules of the survey. We provide more details on the sampling approach in a methods supplement at S1 Appendix. Sample size was determined to estimate trends in programmatic indicators with adequate precision over the planned series of surveys, including estimation of care for childhood illness within five percentage points.

Surveys were adapted from the Liberian Demographic and Health Survey and are provided in an appendix at S1 Survey and administered face-to-face in Liberian Vernacular English and Bassa, the local language in the county. Instruments were translated to Liberian Vernacular English and back-translated to American English to check translation accuracy. Bassa does not have a commonly used written form, but all enumerators were bilingual and trained to administer the survey in both languages. We recruited enumerators who had prior experience administering surveys in this setting, and provided a five-day supplemental training on the questionnaire, its administration, and research ethics with human subjects. Enumerators recorded responses on Android mobile phones and uploaded data regularly to facilitate data quality assurance and enumerator supervision. We discarded data collected by one enumerator because those surveys failed a quality assurance check designed to ensure that enumerators spent enough time surveying each household to obtain accurate information. This resulted in data for 82 children (41 intervention and 41 comparison) being excluded from the follow-up analysis, which was 10.2% of children in that round of surveys. We tested for sensitivity to excluding these observations by re-running our primary analyses including them, and this check is provided in S2 Appendix.

## Variables

Our primary outcome of interest was receipt of care from a qualified provider if a child has suspected diarrhea, malaria, or acute respiratory infection (ARI) in the two weeks preceding the survey. Suspected malaria and diarrhea were defined by maternal report of fever and diarrhea, respectively. Suspected ARI was defined as maternal report of cough plus fast or difficult breathing [17]. Qualified providers were defined as facility-based providers, community health assistants, or general community health volunteers (GCHVs, a cadre of community health workers that preceded the NCHAP). Other non-qualified provider types included drugstores, black baggers (informal drug dispensers), and traditional providers. For our primary outcome, we assessed care for any of the three conditions; we analyzed each condition separately as secondary outcomes. As additional secondary outcomes, we assessed quality of care using maternal report of oral rehydration therapy for diarrhea and rapid diagnostic testing for malaria. We also assessed what providers were sources of care as secondary outcomes to determine whether care providers changed when the community health assistant program was implemented and to directly assess causal mechanisms.

Our exposure of interest was residence in implementation areas rather than comparison areas in the period after implementation. We treated the following variables as potential confounders: child's gender, child's age in months (continuous), maternal age in years (continuous), maternal education (categorized as none, some primary school, or completed at least primary school), whether the mother preferred to complete the survey in English or Bassa, whether the child was reported as having only one illness or more, the number of children under age five residing in the household (continuous), whether the household resides in a community with a primarily agricultural or mining economy, distance to the nearest health facility (log-transformed continuous variables), and quintiles of household wealth. We calculated household wealth using the Filmer-Pritchett approach, which uses the first principal component in a principal component analysis of responses to a household asset index [18].

## Statistical analyses

We used a difference-in-differences approach to estimate whether before-to-after implementation period changes were greater in the intervention than comparison areas. In the unadjusted analysis, we fit a linear probability model in which the outcome was regressed on indicator

variables for intervention versus comparison area, year, and their interaction. The difference-in-differences can then be obtained as the coefficient on the interaction intervention-by-year interaction term. We accounted for potential changes in the composition of the population over time by using inverse probability of treatment weighting (IPTW) to balance all of the covariates listed above except for residence in a mining community across all four time-by-intervention groups. We assessed balance using standard IPTW diagnostics (see S1 Appendix) and then fit an IPT-weighted linear probability model that otherwise had the same form as the unadjusted analysis [12, 19–21].

We assessed changes in provider types using survey design-adjusted tests for differences in proportions. We assessed equity in program outcomes by examining before-to-after changes in childhood disease care in the intervention areas, stratified by distance from the nearest health facility (dichotomized as 5–9.9km versus 10 or more km based on prior research on distance as a barrier in a similar Liberian setting [22]), household wealth (above or below the median), and maternal education (none versus some).

We conducted several sensitivity analyses (see S2 Appendix). First, we could not include residence in mining rather than agricultural communities in the IPTW-adjusted models due to the small number of mining communities, but we conducted supplemental analyses restricted only to agricultural communities. Second, we accounted for confounders using regression adjustment rather than IPTW. These analyses were conducted by fitting logistic regression models in which care from a qualified provider was regressed on indicator variables for intervention, year, and their interaction, as well as all of the confounders listed above. We then estimated difference-in-differences using contrasts of predictive margins after regression. Finally, we re-ran analysis including observations we excluded for data quality assurance, as described above.

All analyses incorporated sampling weights and accounted for stratification and clustering by Taylor linearized standard errors. The IPTW propensity score model incorporated sampling weights, so the final IPTW difference-in-differences model balances the weighted distribution of covariates and remains population-representative [23]. Analyses used Stata 17.0 We provide more detail on statistical methods in S1 Appendix, and statistical code is provided in S1 Code.

Participants gave verbal informed consent. Approval for the surveys was provided by the institutional review boards at the University of Liberia (#18-11-140), Liberia Institute for Biomedical Research (#EC/LIBR/012/037), Partners Healthcare (#2013P002480/PHS), and Georgetown University (#2013–1385).

## Results

Among households approached, response rates were 90% in 2018 and 83% in 2019. We included 2538 households in 2018 and 1313 households in 2019. Within eligible households, 95% of listed women participated in 2018 and 98% in 2019; information about 90% of listed children was provided in 2018 and 98% in 2019. No variable had more than 0.6% missing data.

In the analysis sample, we included data on a total of 1351 children under the age of five in 2018 and 721 in 2019. Among children under five, the burden of at least one childhood disease in the 2 weeks prior to the survey was 70.0% within intervention areas at baseline, 49.9% within intervention at follow-up, 64.0% and 65.6% within comparison areas at baseline, and follow-up respectively.

Overall, samples were similar across years and areas prior to inverse probability of treatment weighting (Table 1); however more households in the comparison group were in mining communities in both surveys, fewer surveys were completed in English in 2019 for both

**Table 1. Characteristics of children with fever, acute respiratory infection, or diarrhea in the two weeks before the survey (prior to inverse probability of treatment weighting).**

| | Pre-Intervention (n = 894) | | Post-Intervention (n = 397) | |
|---|---|---|---|---|
| | Intervention (n = 347), % (95% CI) or Mean (95% CI) | Comparison (n = 547), % (95% CI) or Mean (95% CI) | Intervention (n = 214), % (95% CI) or Mean (95% CI) | Comparison (n = 183), % (95% CI) or Mean (95% CI) |
| Child's household located in mining community, % | 0 | 12.8 (6.7, 23.0) | 6.9 (1.8, 23.5) | 11.7 (4.0, 29.4) |
| Child's gender, % female | 47.9 (43.5, 52.3) | 47.3 (43.3, 51.2) | 40.7 (35.6, 46.0) | 47.2 (39.2, 55.3) |
| Child's age in months, mean | 23.6 (22.2, 25.1) | 27.1 (25.9, 28.2) | 24.2 (22.2, 26.3) | 25.6 (23.4, 27.8) |
| Mother's education, % | | | | |
| No education | 69.2 (63.2, 74.6) | 61.9 (57.1, 66.5) | 73.9 (66.4, 80.2) | 70.4 (61.6, 77.9) |
| Some primary education | 24.9 (20.5, 29.8) | 26.4 (22.4, 30.3) | 21.8 (15.8, 29.2) | 18.5 (12.5, 26.4) |
| Completed primary or higher | 5.9 (3.5, 9.9) | 12.0 (9.4, 15.2) | 4.4 (2.3, 8.2) | 11.1 (7.2, 16.9) |
| Mother's survey language, % | | | | |
| English | 45.0 (32.5, 58.2) | 41.4 (35.2, 47.8) | 20.6 (12.4, 32.4) | 29.9 (18.7, 44.0) |
| Bassa | 55.0 (41.8, 67.6) | 58.7 (52.2, 64.9) | 79.4 (67.6, 87.6) | 70.1 (56.0, 81.3) |
| Mother married or cohabitating, % | 82.3 (74.2, 88.3) | 85.7 (82.1, 88.6) | 91.2 (84.8, 95.1) | 93.7 (88.2, 96.7) |
| Number of illnesses per child in the past 2 wk, % | | | | |
| 1 illness | 38.7 (33.8, 43.8) | 44.4 (39.9, 49.0) | 56.2 (47.0, 65.0) | 66.9 (58.9, 74.0) |
| ≥ 2 illnesses | 61.3 (56.2, 66.2) | 55.6 (51.0, 60.2) | 43.8 (35.0, 53.0) | 33.1 (26.0, 41.1) |
| Child's household distance from facility in km, mean | 15.9 (13.0, 18.9) | 14.6 (12.5, 16.6) | 15.6 (14.0, 17.2) | 14.2 (10.9, 17.6) |
| Child's household wealth index score, mean | -0.18 (-0.52, 0.17) | -0.03 (-0.29, 0.23) | -0.58 (-0.76, -0.41) | 0.09 (-0.25, 0.43) |
| Mother's no. of children, mean | | | | |
| Aged < 5 y | 1.6 (1.5, 1.7) | 1.5 (1.5, 1.6) | 1.5 (1.4, 1.6) | 1.4 (1.2, 1.5) |
| Aged < 1 y | 0.5 (0.4, 0.5) | 0.3 (0.3, 0.4) | 0.4 (0.3, 0.5) | 0.3 (0.2, 0.4) |
| Mother's age, y, mean | 29.0 (28.3, 29.8) | 28.5 (27.9, 29.2) | 28.8 (27.8, 29.8) | 29.5 (28.2, 30.8) |

Note: The sample sizes in this table are unweighted. The percentages were weighted by sampling weights but not by inverse probability of treatment weighting. Thus, they should be interpreted as population-representative distributions of each variable prior to balancing by IPTW. A version of this table showing balance after applying inverse probability of treatment weighting is available in S1 Appendix.

intervention and control, fewer illnesses were reported for children in 2019 for both intervention and control, and wealth was higher in the control group in 2019. In all groups, IPT weighting produced approximate balance, as measured by standardized differences. We present full IPT weighting balance diagnostics and an IPT-weighted version in S1 Appendix.

## Changes in childhood care and care source

In IPT-weighted models, care for all three illnesses from a qualified provider increased by 56.4 percentage points (95% CI: 45.7, 67.1) in intervention areas and decreased by 3.9 percentage points (95% CI: -15.2, 7.5) in comparison areas, a difference-in-differences of 60.3 percentage points (95% CI: 44.7, 76.0; p < 0.001) (Table 2; Fig 1). The disease-specific difference-in-differences between intervention and control areas were 64.5% (95% CI: 47.6, 81.4; p < 0.001) for fever, 74.3% (95% CI: 58.5, 90.0; p < 0.001) for diarrhea, and 48.2% (95% CI: 17.5, 79.0; p = 0.002) for ARI. The difference-in-differences for treatment of diarrhea with oral rehydration therapy (ORT) and receipt of malaria rapid diagnostic tests (RDT) for fever were 37.6%

**Table 2. Difference-in-differences in childhood illness care.**

| Care-seeking from qualified provider | Sample Size (unweighted) | | Unadjusted Model | | Inverse Probability of Treatment Weighted Model | |
|---|---|---|---|---|---|---|
| | Pre | Post | DiD % (95% CI) | p | DiD % (95% CI) | p |
| Any illness | 894 | 397 | 55.2 (41.2, 69.3) | <0.001 | 60.3 (44.7, 76.0) | <0.001 |
| Fever | 691 | 293 | 60.7 (45.1, 76.2) | <0.001 | 64.5 (47.6, 81.4) | <0.001 |
| Diarrhea | 590 | 215 | 69.8 (54.5, 85.1) | <0.001 | 74.3 (58.5, 90.0) | <0.001 |
| Acute Respiratory Illness | 295 | 79 | 39.1 (10.9, 67.2) | 0.007 | 48.2 (17.5, 79.0) | 0.002 |
| Oral rehydration therapy for diarrhea | 586 | 215 | 32.9 (15.0, 50.7) | <0.001 | 37.6 (19.5, 55.8) | <0.001 |
| Rapid diagnostic test for fever | 686 | 293 | 41.9 (26.0, 57.8) | <0.001 | 38.5 (19.9, 57.0) | <0.001 |

(95% CI: 19.5, 55.8; p < 0.001) and 38.5% (95% CI: 19.9, 57.0; P < 0.001), respectively. The regression-adjusted and unadjusted models showed similar results.

Results did not change meaningfully in any of the sensitivity analyses (see S2 Appendix).

There was a significant increase in the proportion of children receiving care from community health assistants (0% to 81.6%; p<0.001) from baseline to follow-up in the intervention areas (Fig 2). In the comparison areas, care from a community health assistant also increased from 0% to 2.0% but this was not significant (p = 0.154). The increase in care seeking in the intervention area was accompanied by decreases in care seeking to non-qualified providers including drugstores and informal providers, with similar changes not seen in the comparison areas.

## Program equity

At baseline in the intervention areas, care from a qualified provider was 14.6 percentage points more common for children living within 10km from a health facility than those farther than 10km. Care increased by 55.3 percentage points (95%CI 44.2, 66.3) after implementation in the farther communities and 63.0 percentage points (95%CI 48.9, 77.2) in the closer communities. The 7.7 percentage point widening of the gap was not significant (95%CI -10.2, 25.7, p = 0.395) (Fig 3). There were no significant differences in care from a qualified provider at baseline in intervention areas by maternal education or household wealth. After implementation, care by a qualified provider increased by 60.8 percentage points (95%CI 51.2, 70.3) among children whose mothers had no formal education, compared to 44.3 percentage points (95%CI 25.4, 63.3) for children of mothers with some education. The 16.5 percentage point greater improvement was not significant (95%CI -1.9, 34.8, p = 0.078). Care from a qualified provider improved by 57.1 percentage points (95%CI 44.3, 69.9) after implementation for children in households below median wealth and 59.1 (95%CI 46.8, 71.3) for children in households above median wealth. The 1.9 percentage point difference in improvements was not significant (95CI CI -14.4, 18.1, p = 0.816)

## Discussion

This study finds evidence that Liberia's National CHA Program led to substantial improvements in care coverage for childhood fever, diarrhea, and ARI. Care sought from qualified providers increased by 60 percentage points more in intervention areas than comparison areas. Receipt of essential elements of care—ORT for diarrhea and RDT-based diagnoses for malaria —also increased by about 40 percentage points more than in comparison areas. In support of the role for the CHA program in causing these gains, we also found that care shifted predominantly to CHAs and sharply away from traditional providers and unregulated drugstores and

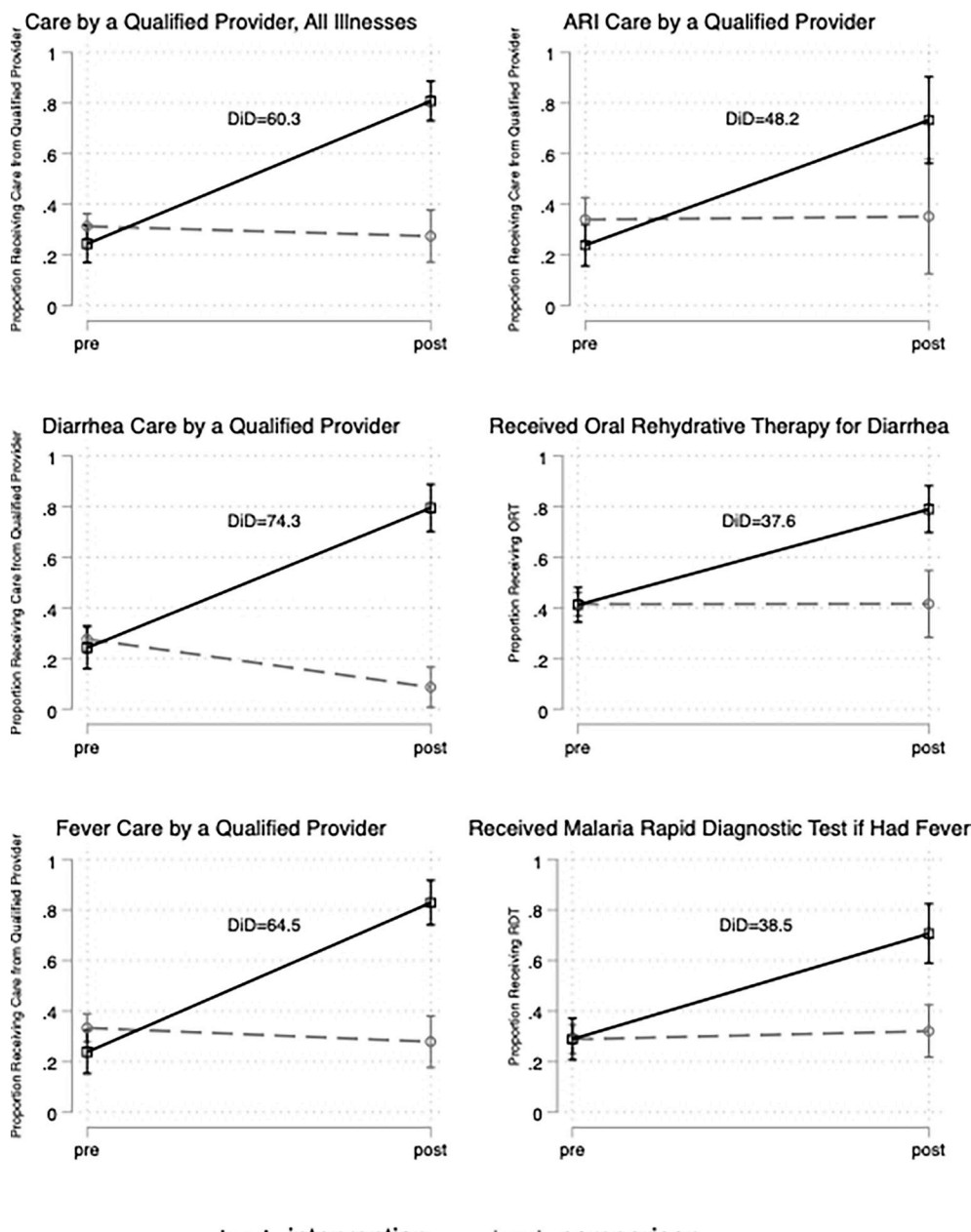

**Fig 1. Differences in before-to-after changes in childhood illness care between intervention and comparison areas.**

informal drug dispensers, as well as a shift away from facility-based providers of about 15 percentage points. The substantial increases in childhood care improvements were seen irrespective of wealth, distance from health facility, or maternal education.

The research design employed in this study enables an inference that the National CHA Program caused the observed improvements. The difference-in-differences framework relies on the assumption that the before-to-after trend observed in the comparison areas represent what would have occurred in the intervention areas but for the intervention [24]. Ideally, this

## Percent of Sick Children Receiving Care from Each Provider Type

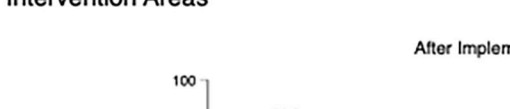

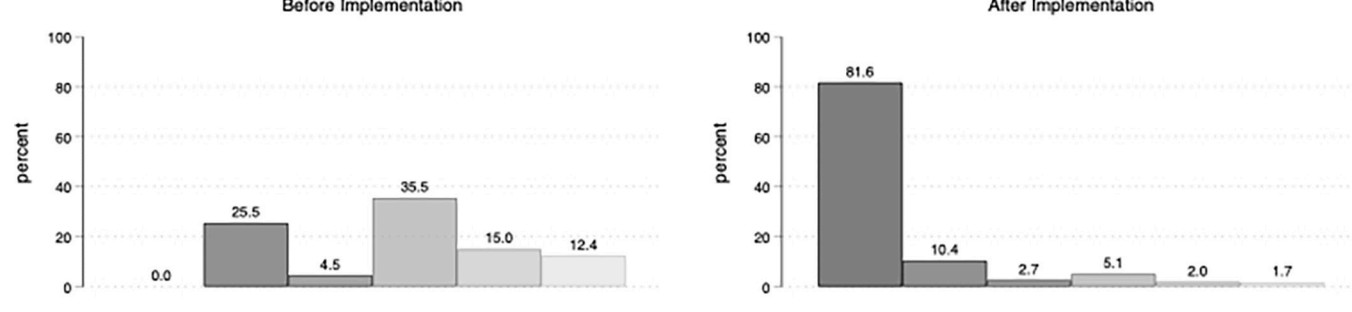

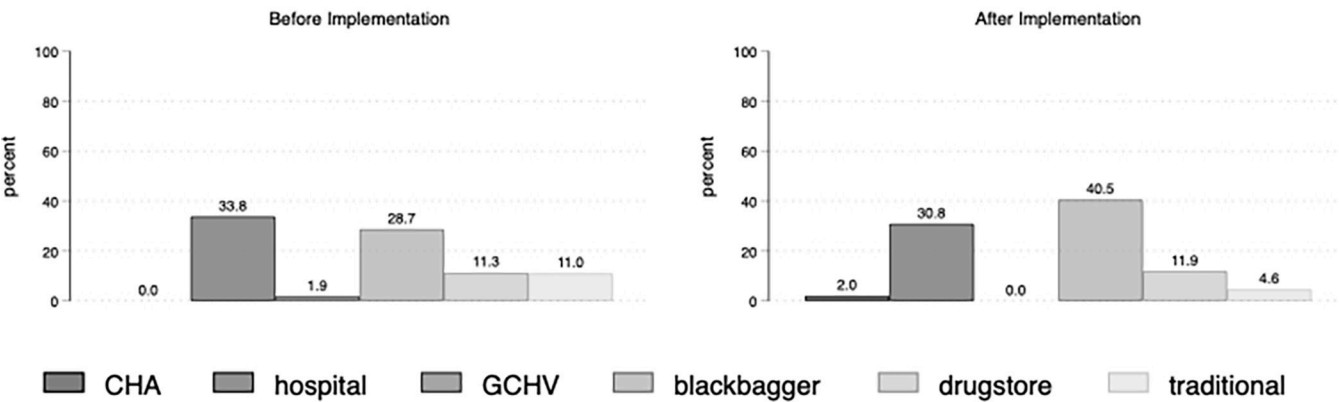

**Fig 2. Percent of sick children receiving care from each provider type.**

assumption can be tested by assessing pre-intervention trends in both areas, but that was precluded by the single pre-intervention measurement that was possible before implementation. However, the inverse probability of care weighting that was also applied mitigated the risk of differing trends by balancing known determinants of childhood illness care across the intervention and comparison areas and time periods [12, 19, 23]. Finally, we could observe the sharp shift toward care from CHAs that occurred shortly after program implementation, and we could not identify any other change that occurred simultaneously and which would be expected to produce such a change.

These findings are broadly consistent with the existing literature on CHW programs. Evaluation of precursor programs to Liberia's National CHA Program found substantial improvements in childhood illness care seeking from a qualified provider, including consistent improvements in care for childhood fever, diarrhea, and acute respiratory illness by qualified providers [11, 12]. Further, Liberian community health assistants have been found to have acceptable levels of knowledge about childhood disease treatment [25]. Finally, a recent analysis of Liberian District Health Information System data found that, in areas served by community health assistants after the launch of the NCHAP program, a little over half of malaria diagnoses had shifted from facilities to community health assistants, 95% of diagnoses were confirmed with RDT or microscopy, and diagnosis appeared to be sustained during the first nine months of the covid-19 pandemic [26]. While our evaluation only focuses on one portion

## Care from a Qualified Provider in Intervention Areas

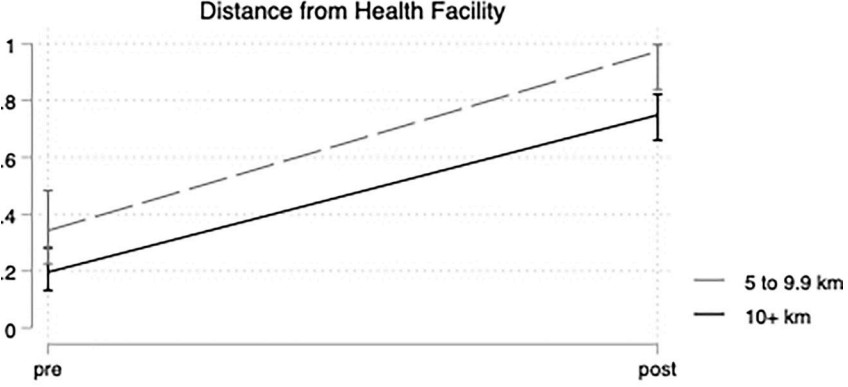

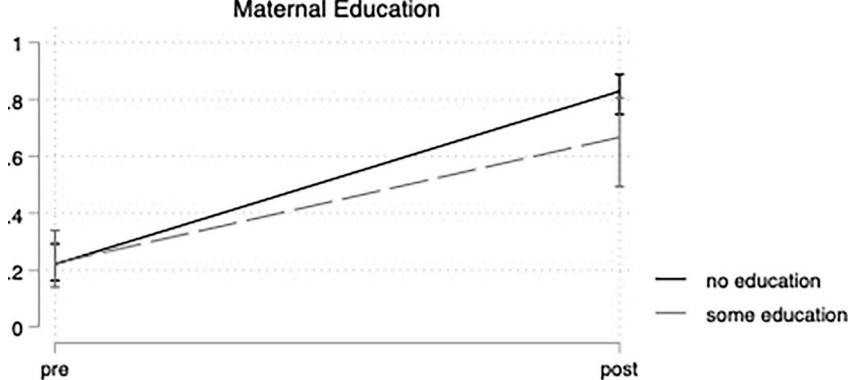

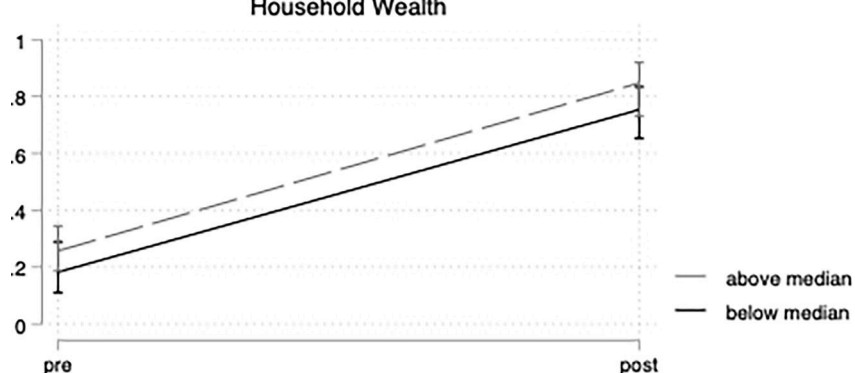

**Fig 3. Improvements in childhood illness care by key dimensions of equity in implementation areas.**

of the country, Liberia's national program now covers over 80% of communities greater than 5km from health facilities, and we would expect similar impact across the national program [14].

Our findings add to a body of literature indicating that CHWs are an important aspect of rural primary care systems in increasing access to and delivery of evidence-based care for children under-5. Systematic reviews, however, have differed about the strength of evidence that community health worker programs produce meaningful improvements in child health

outcomes [10, 27]. These differences likely result from variation in both the package of interventions and delivery model employed by different community health worker programs and the context in which they are implemented. Perry et al. showed that the most effective CHW programs employ staff who are rooted in the communities they serve while also integrated into the larger health system, with salaries and supervision and supplies and able to successfully refer patients who need care beyond what the CHWs can provide [28]. Further, reasonable concerns are often raised about the difficulty of taking CHW programs to national scale [29]. To that end, our results join evidence from other countries that public sector-led national programs can achieve can meaningfully improve care for childhood illnesses when they are able to induce demand for community-based services [30–34].

Our results also corroborate existing literature about the equity and reach of CHW programs. We found improvements were similarly large among groups that have been traditionally disadvantaged as those that are relatively advantaged–such as those further from health facilities, with lower wealth, and with lower education. We interpret these results as suggesting equitable reach and acceptability of the program. Prior research has generally found improvements in child health outcomes across maternal education, household wealth, and rural versus urban residence [35]. However, so far as we could identify, prior research has not examined program equity by degree of remoteness within rural areas, even though distance is a well-known determinant of health outcomes that CHW programs are designed to address and there tends to be worse outcomes at greater relative distances from health facilities, even within populations that are fully rural [6, 36]. It is particularly important to assess equity by distance because CHW supervision, supply chains, and referral networks all become more difficult in more remote communities.

This study has limitations. We lacked statistical power to directly assess changes in childhood mortality as a result of the community health assistant program, though we would expect that increases in coverage of evidence-based interventions known to reduce amenable under-5 mortality would result in decreases in mortality, based on prior literature [37]. We are executing a subsequent study in 2022 that is powered to assess child mortality. This study will also be able to assess whether the gains reported in this manuscript are sustained over time. We discarded one enumerator's data because of concerns about data quality. However, because data were discarded evenly from both intervention and comparison areas, we do not have reason to believe it caused any bias and, in a sensitivity analysis, results did not change meaningfully when that enumerator's data were included. Finally, our study is subject to the recall, reporting, and recording errors that can occur in survey research. We reduced these risks by using standardized Demographic and Health Survey items, building reporting validation into the data collection program, and through training and supervision of enumerators. Parental report of malaria diagnostic testing, in particular, has previously been found to be less than 70% sensitive, which may explain differences between improved fever care and RDT use [38]. Social desirability may have encouraged respondents to decrease reporting of use of unqualified providers, but this would not explain the magnitude of overall increases in coverage we observed. Data entry errors are inevitable, but in previous assessment of our data systems, we found recording error rates below 2% [39].

Overall, this study provides evidence that Liberia's National Community Health Assistant Program has resulted in significant improvements in the care of common child health illnesses, which are estimated to cause about half of under-five deaths in Liberia [40]. Not only did care coverage increase dramatically in intervention areas while it remained approximately constant in comparison areas, but the receipt of essential elements of care also improved along with a shift from unqualified providers mainly to community health assistants. All of these findings suggest that the Liberian National Community Health Assistant Program as

implemented has been successful in strengthening the systems of care and reinforces the value of well-designed public-sector community health worker programs to expand the delivery of equitable, quality care.

## Supporting information

**S1 Appendix. Methods supplement.**
(DOCX)

**S2 Appendix. Sensitivity analyses.**
(DOCX)

**S1 Survey. Survey instrument.**
(DOCX)

**S1 Code. Replication statistical code.**
(RTF)

**S1 Text. Inclusivity in global research questionnaire.**
(DOCX)

## Acknowledgments

We would like to thank the Last Mile Health Liberia programs and monitoring, evaluation and learning teams for their commitment to supporting the Liberia MOH and Grand Bassa County Health Team in implementing the program and in conduct of the household survey, and our External Advisory Committee for their inputs into study design and interpretation. We also would like to thank the women who responded to the questions, and the CHAs for their remarkable commitment to delivery of quality care to the members of their community.

## Author Contributions

**Conceptualization:** Emily White, Savior Mendin, Mark J. Siedner, John D. Kraemer, Lisa R. Hirschhorn.

**Data curation:** Emily White, Robert Karlay, John D. Kraemer.

**Formal analysis:** Emily White, John D. Kraemer.

**Investigation:** Emily White, Savior Mendin, Featha R. Kolubah, Ben Grant, George P. Jacobs, Marion Subah, Mark J. Siedner, John D. Kraemer, Lisa R. Hirschhorn.

**Methodology:** Emily White, Mark J. Siedner, John D. Kraemer, Lisa R. Hirschhorn.

**Project administration:** Savior Mendin, Featha R. Kolubah, Ben Grant, George P. Jacobs, Marion Subah, John D. Kraemer, Lisa R. Hirschhorn.

**Supervision:** Savior Mendin, Marion Subah, John D. Kraemer, Lisa R. Hirschhorn.

**Validation:** Emily White, Robert Karlay.

**Visualization:** Emily White, John D. Kraemer.

**Writing – original draft:** Emily White, Savior Mendin, John D. Kraemer, Lisa R. Hirschhorn.

**Writing – review & editing:** Featha R. Kolubah, Robert Karlay, Ben Grant, George P. Jacobs, Marion Subah, Mark J. Siedner.

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
