## [Decision Letter · Decision Letter 0]

7 Feb 2022

PGPH-D-21-01019

Impact of the Liberian National Community Health Assistant Program on Childhood Illness Treatment in Grand Bassa County, Liberia: A Difference-in-Differences Analysis of Population-Based Data

Dear Dr. Kraemer,

Thank you for submitting your manuscript to PLOS Global Public Health. After careful consideration, we feel that it has merit but does not fully meet PLOS Global Public Health’s publication criteria as it currently stands. Therefore, we invite you to submit a revised version of the manuscript that addresses the points raised during the review process.

We look forward to receiving your revised manuscript.

Kind regards,

David Musoke, PhD

Academic Editor

Journal Requirements:

1. Please include a complete copy of PLOS’ questionnaire on inclusivity in global research in your revised manuscript. Our policy for research in this area aims to improve transparency in the reporting of research performed outside of researchers’ own country or community. The policy applies to researchers who have travelled to a different country to conduct research, research with Indigenous populations or their lands, and research on cultural artefacts. The questionnaire can also be requested at the journal’s discretion for any other submissions, even if these conditions are not met.  Please find more information on the policy and a link to download a blank copy of the questionnaire here: https://journals.plos.org/plosone/s/best-practices-in-research-reporting. Please upload a completed version of your questionnaire as Supporting Information when you resubmit your manuscript.

2. We have noticed that you have uploaded supporting information but you have not included a list of legends.  Please add a full list of legends for all supporting information files (including figures, table and data files) after the references list. 

3. In the online submission form, you indicated that "De-identified data are available from the authors upon reasonable request.". All PLOS journals now require all data underlying the findings described in their manuscript to be freely available to other researchers, either 1. In a public repository, 2. Within the manuscript itself, or 3. Uploaded as supplementary information.

4. Please ensure that the details in the Funding Information is the same with Financial Disclosure Statement.

Additional Editor Comments (if provided):

The reviewers raise some key concerns that need to be satisfactorily addressed.

Reviewers' comments:

Reviewer's Responses to Questions

**Comments to the Author**

1. Does this manuscript meet PLOS Global Public Health’s publication criteria? Is the manuscript technically sound, and do the data support the conclusions? The manuscript must describe methodologically and ethically rigorous research with conclusions that are appropriately drawn based on the data presented.

Reviewer #1: No

Reviewer #2: Partly

2. Has the statistical analysis been performed appropriately and rigorously?

Reviewer #1: No

Reviewer #2: Yes

3. Have the authors made all data underlying the findings in their manuscript fully available (please refer to the Data Availability Statement at the start of the manuscript PDF file)?

Reviewer #1: No

Reviewer #2: Yes

4. Is the manuscript presented in an intelligible fashion and written in standard English?

Reviewer #1: No

Reviewer #2: Yes

5. Review Comments to the Author

Reviewer #1: 1. Infant survival is an important investment that can be made by any health system care delivery. A 15% reduction in Under-five mortality recorded in the country under review is justification warranting health system strengthening to sustain gains recorded. The observed weaknesses in the domain of reinforcing and enabling factors is accountable for poor health outcomes for children of under-five years in the country and training community health workers through the implementation of a pilot community-based training program is considered a strategy of overcoming system weaknesses.

2. The title needs modification; “Impact of the Liberian National Community Health Assistant Program on Childhood Illness-Treatment outcomes in Grand Bassa County, Liberia”

3. The research questions may be better written as; Line 119 should read; To what extent would the intervention programme produce changes in child health treatment outcomes between implementation area over control?

4. Overall Observations: There is observed lack of recourse to any theoretical clarifications underpinning health promotion principles and public health principles of prevention and control in the developing the programme and implementation. This translates to a superficial study devoid of theoretical foundation.

Since the problem phenomenon has predominantly two sources of challenges, strategic planning should evolve from diagnostic considerations of the reinforcing and enabling factors underpinning the problem phenomenon. In the process of designing the study, there are no evidence of applying theoretical resources that would guide elucidation of proof of concept. This grossly overlooked application of public health science in the planning and implementation of the study constitute weakness this type of study cannot afford. Secondly, personal-level dispositions of the target population for the intervention, the CHW, have not been considered. These observed lack of theoretical grounding makes the study unscientific and flawed.

In such intervention studies, behaviour models and theories play significant role in providing theoretical-grounding to accommodate predisposing, Reinforcing and Enabling factor considered to be responsible for the problem phenomenon so that the programme developed is based on adequate appropriate diagnostic principles and not superficially assuming that a training package would address the issues responsible for the problems appropriately.

-Statistical analysis is completely flawed. In such pre- and post-test intervention programmes, scores are generated for the target samples (Experimental group and Control group), this was not the case here, rather frequency distribution was used, whereas a paired-sample t-test is most appropriate with a computation of Mean-based Effect Size to determine the magnitude or impact of the intervention.

5. If the authors are able to address the lack of theoretical and conceptual resources as basis for developing the intervention programme and data analysis, then the weaknesses in the study would have been ameliorated.

Reviewer #2: 1. Abstract, paragraph 2. The statement “we measured before-to-after changes in childhood treatment from qualified providers” is too broad. This sentence does not give any indication about what issues related to childhood are being addressed. Suggest you modify to “… changes in treatment of common childhood illnesses from qualified providers…”

2. Abstract, paragraph 2. It is not informative to have percentage increase in your primary outcome in the abstract without any indication of the scale or denominator of your population. It could be an increase among 10 children or among 100,000 people. Please mention the total number of participants in your surveys.

3. Introduction, line 78. Please clarify that this is “all-cause under five mortality”

4. Introduction, line 82. Please either use the standardized term “all-cause under-five mortality”, or specify what causes of death you describing (e.g. only infectious disease deaths).

5. Introduction, line 92. Describing this study as “proof of principle” suggests that you’re conducting a very controlled efficacy study, rather than what I assume is a effectiveness study exploring the impact of the program in a more routine implementation setting. Please clarify.

6. Introduction lines 94-99. Further to my previous comment, if there are already two published evaluations of this project, why is this analysis considered the proof of principle? How does this paper add to what’s already published, or is each county just getting a separate publication? Is this a larger scale evaluation with more data / areas included? Used a different implementation / support module? MoH led rather than partner agency?

7. Introduction line 106-7. This suggests your current analysis will evaluate the full national program. Why was this not done, for example using a stepped wedge to account for the phased roll-out of the program?

8. Introduction, line 114-5. Need to also mention the key DiD assumption that intervention & comparison areas are assumed to be otherwise similar.

9. Methods. Some additional information is required to clarify the county mapping and sampling procedures.

a. Did you conduct a complete census of every household in the study counties by sending teams out to visit all houses? Or get estimated number of households from local leaders or administration, or use existing government national census data?

b. Please present your sample size assumptions to confirm that you had “adequate precision over the planned series of surveys”

c. What was the modified random walk? Please include a couple of sentences explaining how this was done.

d. What indicators did you use to ensure intervention and comparator districts were comparable on baseline health indicators? Did you also include these indicators in your IPW or multivariable models?

e. How did you sample based on PPS unless you have a complete census?

10. Methods, line 178. I’m curious why your primary outcome of interest is receiving treatment from a qualified provider, rather than just seeing care from a qualified provider. Particularly for children with suspected malaria or suspected pneumonia, treatment should only be provided if the child meets the criteria stated in national guidelines (e.g. positive malaria rapid test). Does this mean that children seeking care but who didn’t receive any drugs are excluded from the analysis?

11. Results, line 241. Does the 2538 households in 2018 include participants from counties that weren’t included in the final DiD analysis? If it was just the counties included in the analysis, can you explain why the sample size is so much smaller in 2019?

12. Discussion, line 313. If you are proposing that your results can be interpreted causally, I suggest providing further justification from the evaluation literature that your analysis approach and the implementation environment are suitable to make these causal conclusions.

13. Discussion lines 314-315. Is this availability of ORS and RDTs at trained health providers (e.g. lack of stock outs) or receipt of these commodities by eligible individuals?

14. Discussion, line 356. I find the equity by distance principle a little hard to understand, since the CHA program is presumably targeted to the communities furthest from health facilities. Did you look at distance from a qualified care provider after the CHA program was rolled out, or before? Is this question actually asking about whether CHAs were targeted to the most in-need communities?

15. Discussion, line 377. Suggest change “resulted in” to “associated with” unless you strengthen your justification that this is a causal effect analysis.

16. Discussion, line 380-1. I didn’t see any data on quality of care, only that individuals were attending qualified rather than unqualified health providers. Did you assess the quality of care of CHAs compared to a sample of the unqualified providers?

17. In the discussion, it would be interested to hear if you anticipate these levels of usage of CHAs to remain stable over time, since the second surveys was only 5 months after the intervention roll out, do you anticipate these levels of usage of CHAs to remain stable over time? Is there any experience from the earlier pilots in Liberia?

18. Figure 1. Please add labels to the y-axes.

19. Figure 2. Percent of children receiving what treatment? Or do you mean seeking care from a qualified health provider in the last 2 weeks?

20. Figure 3. As previous comment. Treatment of what?

21. Supplement 1. Sampling. You mention in the main article using PPS, but here say it was a simple random sample. I am also still unclear about how you did PPS or the simulations mentioned here without further information on whether county-wide census was performed. Furthermore, if you had a full census for the sampling, why did you then use random walk to select households – which may result in some geographic biases – rather than a random selection from your census data?.

22. Supplement 1. IPTW. What is meant by the number of illnesses? Should that have been included in the IPW considering it is independently associated with outcomes relating to treatment?

6. PLOS authors have the option to publish the peer review history of their article (what does this mean?). If published, this will include your full peer review and any attached files.

**Do you want your identity to be public for this peer review?** For information about this choice, including consent withdrawal, please see our Privacy Policy.

Reviewer #1: **Yes: **Professor Nnodimele Atulomah

Reviewer #2: No

---

## [Decision Letter · Decision Letter 1]

2 May 2022

PGPH-D-21-01019R1

Impact of the Liberian National Community Health Assistant Program on Childhood Illness Care in Grand Bassa County, Liberia: A Difference-in-Differences Analysis of Population-Based Data

Dear Dr. Kraemer,

Thank you for submitting your manuscript to PLOS Global Public Health. After careful consideration, we feel that it has merit but does not fully meet PLOS Global Public Health’s publication criteria as it currently stands. Therefore, we invite you to submit a revised version of the manuscript that addresses the points raised during the review process.

We look forward to receiving your revised manuscript.

Kind regards,

David Musoke, PhD

Academic Editor

Journal Requirements:

Additional Editor Comments (if provided):

Reviewer 1 still raising some pertinent concerns that need to be addressed by the authors.

Reviewers' comments:

Reviewer's Responses to Questions

**Comments to the Author**

1. If the authors have adequately addressed your comments raised in a previous round of review and you feel that this manuscript is now acceptable for publication, you may indicate that here to bypass the “Comments to the Author” section, enter your conflict of interest statement in the “Confidential to Editor” section, and submit your "Accept" recommendation.

Reviewer #1: (No Response)

Reviewer #2: All comments have been addressed

Reviewer #3: All comments have been addressed

2. Does this manuscript meet PLOS Global Public Health’s publication criteria? Is the manuscript technically sound, and do the data support the conclusions? The manuscript must describe methodologically and ethically rigorous research with conclusions that are appropriately drawn based on the data presented.

Reviewer #1: No

Reviewer #2: Yes

Reviewer #3: Yes

3. Has the statistical analysis been performed appropriately and rigorously?

Reviewer #1: No

Reviewer #2: Yes

Reviewer #3: Yes

4. Have the authors made all data underlying the findings in their manuscript fully available (please refer to the Data Availability Statement at the start of the manuscript PDF file)?

Reviewer #1: No

Reviewer #2: Yes

Reviewer #3: No

5. Is the manuscript presented in an intelligible fashion and written in standard English?

Reviewer #1: No

Reviewer #2: Yes

Reviewer #3: Yes

6. Review Comments to the Author

Reviewer #1: All recommendations suggested in the first round of review to strengthen the manuscript were totally ignored.

The following recommendation where made:

1. The observed weaknesses in the domain of reinforcing and enabling factors is accountable for poor health outcomes for children of under-five years in the country and training community health workers through the implementation of a pilot community-based training program is considered a strategy of overcoming system weaknesses. THESE WERE NOT CONSIDERED IMPORTANT SOURCE OF THE PROBLEM TO BE ESTABLISHED AS FOUNDATIONAL PRINCIPLES TO JUSTIFY THE INTERVENTION.

2. The title needs modification; “Impact of the Liberian National Community Health Assistant Program on Childhood Illness-Treatment outcomes in Grand Bassa County, Liberia” NO CHANGE IN THE TITLE RECOMMENDED, WAS MADE.

3. To what extent would the intervention programme produce changes in child health treatment outcomes between implementation area over control? NOT EFFECTED.

4. Statistical analysis is completely flawed. In such pre- and post-test intervention programmes, scores are generated for the target samples (Experimental group and Control group), this was not the case here, rather frequency distribution was used, whereas a paired-sample t-test is most appropriate with a computation of Mean-based Effect Size to determine the magnitude or impact of the intervention. Currently, global best practices for before-and-after intervention would quantify changes as Effect Size, which was recommended but not considered.

Reviewer #2: Thank you for your detailed response to my previous comments. All issues have been addressed satisfactorily.

Reviewer #3: Title: Impact of the Liberian National Community Health Assistant Program on Childhood Illness Care in Grand Bassa County, Liberia: A Difference-in-Differences Analysis of Population-Based Data.

This topic is interesting and the method used is what capture my attention to reading this manuscript. It also shows the pre and post-intervention levels of the national health system of Liberia, showing how they have measured the impact and the consequent focus on the future sustainability of the program. Therefore, this paper is worth publishing.

The only thing to improve on is for the author to be detailed in the main body of the paper on how they derive the various statistics that were presented in this paper, especially the part of the difference-in-difference analysis.

Thank you.

7. PLOS authors have the option to publish the peer review history of their article (what does this mean?). If published, this will include your full peer review and any attached files.

**Do you want your identity to be public for this peer review?** For information about this choice, including consent withdrawal, please see our Privacy Policy.

Reviewer #1: No

Reviewer #2: No

Reviewer #3: **Yes: **Damola Bakare

---

## [Editor Report · Decision Letter 2]

1 Jun 2022

Impact of the Liberian National Community Health Assistant Program on Childhood Illness Care in Grand Bassa County, Liberia

PGPH-D-21-01019R2

Dear Prof. Kraemer,

We are pleased to inform you that your manuscript 'Impact of the Liberian National Community Health Assistant Program on Childhood Illness Care in Grand Bassa County, Liberia' has been provisionally accepted for publication in PLOS Global Public Health.

Best regards,

David Musoke, PhD

Academic Editor